# Investigation of Adverse Events Experienced by Healthcare Workers following Immunization with Homologous or Heterologous COVID-19 Booster Vaccinations

**DOI:** 10.3390/vaccines10111869

**Published:** 2022-11-04

**Authors:** Yunhua Wei, Yan Wang, Jian Liu, Yan Zha, Yuqi Yang, Ni Li, Yalin Zhou, Jinli Zhu, Neil Roberts, Lin Liu, Yaying Li

**Affiliations:** 1Department of Nuclear Medicine, Guizhou Provincial People’s Hospital, Affiliated Hospital of Guizhou University, Guiyang 550002, China; 2Department of Obstetrics and Gynecology, Guizhou Provincial Staff Hospital, Huaxi Branch Affiliated to Guizhou Provincial People’s Hospital, Guiyang 550003, China; 3Department of Neurosurgery, Guizhou Provincial People’s Hospital, Affiliated Hospital of Guizhou University, Guiyang 550002, China; 4Department of Nephrology, Guizhou Provincial People’s Hospital, Affiliated Hospital of Guizhou University, Guiyang 550002, China; 5Department of Medical Care, Guizhou Provincial People’s Hospital, Affiliated Hospital of Guizhou University, Guiyang 550002, China; 6Department of Traditional Chinese Medicine, Guizhou Provincial Staff Hospital, Huaxi Branch Affiliated to Guizhou Provincial People’s Hospital, Guiyang 550003, China; 7School of Clinical Sciences, The Queen’s Medical Research Institute (QMRI), University of Edinburgh, Edinburgh EH8 9YL, UK; 8Department of Respiratory and Critical Care Medicine, NHC Key Laboratory of Pulmonary Immunological Diseases, Guizhou Provincial People’s Hospital, Affiliated Hospital of Guizhou University, Guiyang 550002, China

**Keywords:** adverse events following immunization (AEFI), booster COVID-19 vaccination, homologous, heterologous

## Abstract

Objective: A comparative analysis was performed to investigate the potential risk factors of Adverse Events Following Immunization (AEFI) after receiving different booster vaccines. Methods: From 18 January 2021 to 21 January 2022, the Health Care Workers (HCWs) of Guizhou Provincial Staff Hospital (Guizhou Province, China) who received a third Booster vaccine, that was either homologous (i.e., (i) a total of three doses of Vero cell vaccine or (ii) three doses of CHO cell vaccine) or (iii) heterologous with two first doses of Vero cell vaccine, being either CHO cell vaccine or adenovirus type-5 (Ad5) vectored COVID-19 vaccine, were asked to complete a self-report questionnaire form to provide information on any AEFI that may have occurred in the first 3 days after vaccination with the booster. The frequency of AEFI corresponding to the three different booster vaccines was compared, and the risk factors for predicting AEFI were determined by multivariate logistic regression analysis. Results: Of the 904 HCWs who completed the survey, 792 met the inclusion criteria. The rates of AEFI were 9.8% (62/635) in the homologous Vero cell booster group, 17.3% (13/75) in the homologous CHO cell booster group, and 20.7% (17/82) in the heterologous mixed vaccines booster group, and the rates were significantly different (χ^2^ = 11.5, *p* = 0.004) between the three groups of vaccines. Multivariate logistic regression analysis showed that: (1) compared to the homologous Vero cell booster group, the risk of AEFI was about 2.1 times higher (OR = 2.095, 95% CI: 1.056–4.157, *p* = 0.034) in the CHO cell booster group and 2.5 times higher (OR = 2.476, 95% CI: 1.352–4.533, *p* = 0.003) in the mixed vaccines group; (2) the odds for women experiencing AEFI were about 2.8 times higher (OR = 2.792, 95% CI: 1.407–5.543, *p* = 0.003) than men; and (3) compared to the non-frontline HCWs, the risk of AEFI was about 2.6 times higher (OR = 2.648, 95% CI: 1.473–4.760, *p* = 0.001) in the doctors. Conclusion: The AEFI in all three booster groups are acceptable, and serious adverse events are rare. The risk of AEFI was higher in doctors, which may be related to the high stress during the COVID-19 epidemic. Support from government and non-governmental agencies is important for ensuring the physical and mental health of HCWs.

## 1. Introduction

In China, the initial vaccination protocol of the three licensed coronavirus disease 2019 (COVID-19) vaccines (inactivated COVID-19 vaccine (Vero cell), recombinant novel coronavirus vaccine (CHO cell), and adenovirus type-5 (Ad5) vectored COVID-19 vaccine) are very effective [1,2]. However, breakthrough infections occur and reflect how immunity to Severe Acute Respiratory Syndrome Coronavirus 2 (SARS-CoV-2) decreases with time after receiving two doses of the COVID-19 vaccine [3,4,5]. Furthermore, the emergence of variants such as Delta and Omicron has increased the number of breakthrough infections occurring in the fully vaccinated population [4,5,6] throughout the world. Booster vaccination programs have been launched in countries around the world. However, the data on Adverse Events Following Immunization (AEFI) after vaccination with the booster dose of the homologous or heterologous vaccine in fully vaccinated people are still being compiled, and the risk factors associated with AEFI are not fully known. The results of the present study provide new information on the likelihood of an AEFI occurring following the COVID-19 booster vaccinations for HCWs in Guizhou Provincial Staff Hospital.

Faced with short supplies of COVID-19 vaccines and unforeseen side effects, some countries have adopted the unproven strategy of switching the type of vaccine used for the booster vaccine. A review of initial data led to the suggestion that this approach, born of necessity, may actually be beneficial [7]. Subsequently, multiple clinical trials have shown there to be no significant differences in the number of AEFI following use of a homologous or heterologous booster vaccine [8,9]. However, heterologous vaccination elicited a more durable, broader, and more robust cellular and humoral immunity than homologous vaccination, and provides beneficial protection against the SARS-CoV-2 variants. Furthermore, a booster dose of the vaccine was shown to be highly effective in preventing infection, severe disease in cases of breakthrough infection, hospitalization, or death [10]. Nevertheless, the best combination of initial and booster vaccines for protection against COVID-19 is still to be determined.

Presently, among the publicly available research reports, there are none in which a comparative analysis has been performed of the risk factors of AEFI after receiving a booster dose that was either homologous or heterologous with the first two doses of any of the vaccines. Consequently, a preliminary study has been performed at Guizhou Provincial Staff Hospital for HCWs who initially received inactivated COVID-19 vaccine (Vero cell), recombinant novel coronavirus vaccine (CHO cell), or adenovirus type-5 (Ad5) vectored COVID-19 vaccine [11]. Previous studies have reported [12] that there is reciprocal communication between the brain and the immune system, so that just as psychological states can influence the immune response, so too can activation of the immune system influence the brain and behavior. Infection with SARS-CoV-2 and pandemic-related stress may stimulate the immune system, and there may also be alterations in mood, cognition, and behavior. Similarly, changes in peripheral inflammation induced by vaccination may also be associated with changes in symptoms and underlying neural activity. At the beginning of the epidemic of SARS-CoV-2 (2019–2020), the Guizhou Provincial Staff Hospital was designated by the local government as a special medical institution for the treatment of COVID-19 in Guizhou Province. HCWs worked at the frontline of the response to COVID-19 and during the first outbreak when there was a great lack of knowledge regarding COVID-19. HCWs were highly vulnerable to stress. For example, there have been reports [13] that a significant proportion of HCWs in China have developed psychological symptoms, which may affect many aspects of the functioning of the immune, central nervous, and endocrine systems of the body [14]. HCWs have different pressures depending on their role in the workplace [15], and the relationship between occupational category and AEFI is presently unknown. This information is sought to potentially increase confidence in these populations in receiving a third dose of the vaccine. The main objective of the present study was, therefore, to investigate the AEFI caused by administration of homologous or heterologous COVID-19 booster vaccines in HCWs of Guizhou Provincial Staff Hospital; in particular, AEFI that occurred in HCWs who received a booster dose of either homologous inactivated COVID-19 vaccine (Vero cell); homologous recombinant novel coronavirus vaccine (CHO cell); or a heterologous mixing of vaccines, i.e., two doses of Vero cell followed by a different vaccine (i.e., a booster dose of CHO cell or adenovirus type-5 (Ad5), respectively) were compared and analyzed to determine the potential risk of AEFI in the three vaccination groups.

## 2. Methods

### 2.1. Study Design and Population

The study was approved by the Institutional Review Board of Guizhou Provincial Staff Hospital. A total of 1000 HCWs at Guizhou Provincial Staff Hospital who had received at least one dose of a COVID-19 vaccine between 18 January 2021 and 21 January 2022 were invited to complete a mobile-phone-based questionnaire. Because no personally identifiable information was acquired and no human biospecimens were obtained, the Institutional Review Board waived the necessity for written informed consent from each participant.

The questionnaire included requests for information regarding sex, age, ethnicity, level of education, history of COVID-19 infection, the occupational categories of HCWs (such as doctors, nurses, pharmacists, technicians, or members of logistical staff, etc.), history of allergies to the vaccine, types of vaccinations, doses of vaccinations, symptoms of AEFI within 3 days of vaccination, and whether medical attention was required after administration of the vaccine. If the answer to the last question was yes, then further information was requested with respect to whether the patient received outpatient or inpatient treatment, as well as the details of the treatment.

### 2.2. Vaccine

The initial COVID-19 vaccination protocol in China was as follows: (a) inactivated COVID-19 vaccines (Vero cell) administered as two doses within an interval of 3 to 8 weeks; (b) recombinant protein subunit vaccine (CHO cell) administered as a single dose; or (c) recombinant protein subunit vaccine (CHO cell) administered as two doses within an interval of 4 to 8 weeks. The dose of each vaccine was administered according to the instructions of the manufacturer. In October 2021, China’s health authorities launched a booster vaccination program [16], recommending that people who have received two doses of COVID-19 vaccine can receive the third-dose booster vaccination six months later, which can either be (according to the doctor’s advice and the participant’s choice) Vero cell, CHO cell, orAd5 vaccine.

### 2.3. Inclusion Criteria

The inclusion criteria were HCWs who had received a homologous booster dose of either Vero cell or CHO cell, or who received heterologous mixing vaccines (i.e., Vero cell (two doses) and CHO cell (one dose) or Vero cell (two doses) and Ad5 (one dose) vaccine). The HCWs who completed the questionnaire were, thus, assigned to either (i) the homologous Vero cell Booster group, (ii) the homologous CHO cell booster group, or (iii) the heterologous mixed vaccines booster group.

### 2.4. Exclusion Criteria

Not an HCW;People with a history of COVID-19 infection or vaccine allergies;Women who were pregnant or breastfeeding;People with severe chronic or immunocompromised diseases;People aged <18 years or >60 years.

### 2.5. Statistical Analysis

Statistical analysis was performed using GraphPad PRISM, version 9.0.0 (GraphPad Software, CA, USA), and SPSS, version 26 (IBM Corp., Armonk, NY, USA), and the results were considered significant for *p* < 0.05. Values of categorical variables were recorded as frequencies or percentages and were analyzed using the χ^2^ test or Fisher’s exact test. Summary descriptions were prepared regarding the baseline characteristics; whether a HCW required medical attention after administration of a vaccine; and vaccination reactions for the homologous Vero cell booster group, homologous CHO cell booster group, and heterologous mixed vaccines booster group. The *p*-value was adjusted by Bonferroni correction when the χ^2^ test was applied to compare the rates of AEFI between multiple groups.

AEFI was the outcome variable, and occupational categories were defined as disorderly classification variables (i.e., (a) frontline: (i) doctors or (ii) nurses, or (b) non-frontline: (iii) others), and multivariate logistic regression analysis was performed to explore the relative potential risk of AEFI, and associated factors, for the three different vaccine groups. In particular, univariate analysis was performed to assess potential covariation (including sex, age (stratifying by age groups and occupational category)) and afterwards, an adjustment was made for statistically significant covariates in the multivariate logistic regression models (i.e., forward). The odds ratio (OR) and 95% confidence interval (CI) were calculated for the regression model.

## 3. Results

Of the 1000 HCWs who received the questionnaire, 904 completed a full response, including 792 who met the inclusion criteria, and of whom, 75.4% (597/792) were women and 51.4% (407/792) had a Bachelor’s degree or above. A total of 80.1% (635/792) responses were for HCWs in the homologous Vero cell booster group, 9.5% (75/792) for HCWs in the homologous CHO cell booster group, and 10.4% (82/792) for HCWs in the heterologous mixed vaccines booster group. Overall, 11.6% (92/792) of participants experienced an AEFI during the first 3 days after receiving the booster vaccine, and further details of the responses of the participants can be found in Table 1. Only five people sought help from the outpatient provider, but none underwent any medical treatment and the symptoms of AEFI resolved spontaneously.

There were significant differences in the incidence of AEFI between the three groups, with values of 9.8%, 17.3%, and 20.7% in the homologous Vero cell booster group, homologous CHO cell booster group, and heterologous mixed vaccine booster group (χ^2^ = 11.5, *p* = 0.004), respectively (Figure 1), and with muscle pain/headache (5.2%), pain at the injection site (10.7%), and fever (9.8%) as the corresponding most commonly reported AEFI.

The potential risk of AEFI between the three vaccine groups and the factors associated with AEFI were subsequently computed using an adjusted multivariate logistic regression model, and the results are presented in Table 2. The incidence of AEFI was significantly lower in the Vero cell booster group than in the other two groups, and univariate analysis subsequently showed that relative to the Vero cell booster group, the risk of AEFI was about 1.9 times higher (OR = 1.938, 95% CI: 1.009–3.722, *p* = 0.047) in the CHO cell booster group and 2.4 times higher (OR = 2.417, 95% CI: 1.334–4.381, *p* = 0.004) in the heterologous mixed vaccines booster group. The univariate analysis also showed that the risk of AEFI was about 2.6 times higher (OR = 2.626, 95% CI: 1.368–5.039, *p* = 0.004) in women compared with men. When the professional category of the HCWs was considered, the risk of AEFI was about 2.2 (OR = 2.153, 95% CI: 1.234–3.757, *p* = 0.007) times higher in the doctors compared to the other groups combined, but the risk of AEFI showed no significant differences in the nurses (*p* > 0.05) compared to the other groups combined. Moreover, there were no significant differences in the risk of AEFI depending on age group or ethnicity (*p* > 0.05).

The subsequent application of an adjusted multivariate logistic regression analysis showed that compared to the homologous Vero cell booster group, the risk of AEFI was approximately 2.1 times higher (OR = 2.095, 95% CI: 1.056–4.157, *p* = 0.034) in the homologous CHO cell booster group and 2.5 times higher (OR = 2.476, 95% CI: 1.352–4.533, *p* = 0.003) in the heterologous mixed vaccines group. The risk of AEFI was approximately 2.8 times higher (OR = 2.792, 95% CI: 1.407–5.543, *p* = 0.003) in women compared to men. When the occupational category of the HCWs was considered, relative to the other groups combined, the risk of AEFI was about 2.6 times higher (OR = 2.648, 95% CI: 1.473–4.760, *p* = 0.001) in the doctors, but not significantly different in the nurses (*p* > 0.05).

## 4. Discussion

The occurrence of AEFI in HCWs after receiving the booster dose of homologous inactivated COVID-19 vaccine (Vero cell), homologous recombinant novel coronavirus vaccine (CHO cell), or heterologous mixing vaccines has been investigated. This has shown that the incidence of AEFI was low in all the three booster vaccine groups, and none of the AEFIs were particularly serious. The main AEFI reported were fever, pain at the injection site, and muscle pain/headache, which were consistent with clinical trials of similar booster vaccinations [9,17,18]. Moreover, these symptoms coincide with the normal reaction to other vaccines, such as the trivalent influenza vaccine [19], the whole-cell pertussis vaccine [20], and pneumococcal vaccines [21]. Severe adverse events were rare, and included hypersensitivity, facioplegia, urticaria, and anaphylactic shock [22]. In a meta-analysis [23], the lowest incidence of AEFI was observed in the initial vaccination program for the inactivated Vero cell vaccine (two doses). In the booster vaccination program at Guizhou Provincial Staff Hospital, the incidence of AEFI in the homologous Vero cell booster group was also the lowest compared the homologous CHO cell booster group and the heterologous mixed vaccines booster group. A clinical trial in the UK has confirmed [24] that where a homologous comparator was included, reactogenicity appeared increased in people who received a heterologous boost, but was tolerated. Furthermore, administration of the Ad5 Booster after initial immunization with the Vero cell vaccine appears to enhance immunogenicity over that provided by a Vero cell booster.

The factors associated with AEFI in the three groups of participants receiving the booster vaccine were also analyzed, and being a woman was one of the most significant associated risk factors [25,26]. Consequently, the physical and mental health of women should receive particular attention, and long-term monitoring of women entering new vaccination programs is necessary. In addition, after adjustment in a multivariate logistical regression model, being a doctor was a risk factor for experiencing AEFI, although serious AEFI was rare. The authors of an observational longitudinal study conducted in Spain reported that women, administrative workers, and workers who had been infected by COVID-19 tended to report more reactions to the vaccine [27]. The authors suggested that men (especially physicians) tend to underreport their symptoms after vaccination, thus biasing the results of pharmacovigilance studies. Nevertheless, it is well known that one of the most exposed groups to COVID-19 and its psychosocial consequences is HCWs. Especially during the first outbreak, when there was a lack of knowledge regarding COVID-19, the absence of any effective medicine or vaccine and the scarcity of medical material and human resources at that time made HCWs highly vulnerable to stress.

At the beginning of the epidemic of SARS-CoV-2, the Guizhou Provincial Staff Hospital was designated by the local government as a special medical institution for the treatment of COVID-19 in Guizhou Province. The HCWs at the hospital also suffered very high psychological pressure due to the above reasons. Studies have shown that psychological stress may affect many aspects of the integration of functioning of the immune, central nervous, and endocrine systems in both animals and humans [14,28]. A study of psychoneuroimmunology (PNI) [28] showed that stress could induce significant increases in serum IgA, IgG, IgM, C3c, C4, and acute response protein (AP) in vivo combined with high-stress perception. Therefore, the significant relationships between the stress-induced changes in serum Ig concentrations indicate that the latter are acutely sensitive to the effect of stressors [29]. This is interpreted to suggest, albeit indirectly, that the HCWs were a population with high stress levels. Furthermore, studies in which individuals were exposed to a cold or influenza virus have shown [30] that those under chronic stress are more likely to develop upper respiratory infections. These effects extend to the vaccine response, such that stressed individuals mount a diminished antibody response to vaccination [31]. The field of PNI includes studies in which the links between stress and the immune system are examined. This work is particularly relevant in the context of the COVID-19 pandemic, which combines all the elements of a major stressor; namely, the pandemic is unpredictable; uncontrollable; has generated tremendous fear, loss, and grief; has created social challenges and challenges for government; and has disrupted almost all aspects of daily life. Although research in PIN has primarily focused on stress and other negative psychological states, there is growing recognition that positive psychological states may also modulate the immune system and neuroimmune interactions. Of potential relevance for COVID-19, early studies found that social support was associated with enhanced NK cell activity, as well as with measures of adaptive immune function [32,33,34]. Social support and vaccination have provided more mixed results, and social support has been reported to be associated with a stronger immune response to hepatitis B vaccination in medical students [35,36]. Therefore, during the COVID-19 pandemic, increasing psychological and social support for HCWs pre- and post-vaccination, and promoting the need for psychological crisis interventions for medical staff, focusing on their physical and mental health, is highly recommended.

In the multivariate logistical regression model, the risk of AEFI in the mixed vaccines booster group was higher than in the homologous vaccines booster group, although the AEFI that occurred were not more serious. This indicates that the immunogenicity of the heterologous mixed vaccine booster group was acceptable, and this finding is similar to those reported in other studies [9,10,37]. For example, in the UK COMCOV trial, it was reported that a heterologous booster vaccine can provide greater immunity than a homologous booster vaccine [9,38]. Furthermore, a study conducted in the United States revealed [10] that the use of a homologous booster provides a wide range of immunogenicity responses, and heterologous boosting provided similar or higher levels. Moreover, other studies have confirmed [8,39,40,41] that heterologous boosters were not only effective in preventing serious diseases such as infection, but can also improve resilience to infection with Omicron. In December 2021, the SARS-CoV-2 Omicron variant swiftly overtook the Delta variant to become globally dominant in the COVID-19 pandemic [42]. Protective immunity in vaccines has diminished over time. Epidemiology data on Omicron-associated reinfection and vaccine breakthrough infections suggest that this variant exhibits serious evasion of the immune system. Variant-targeted vaccines are being developed [43], and new data confirm that the administration of a booster dose of COVID-19 vaccine is crucial for generating antibody responses to protect against infection by the Omicron variant, although better therapeutic antibodies are also still needed to protect against this and future variants [42]. Nevertheless, the vaccines might not be free from adverse effects that may remain undetectable in clinical trials, so the evaluation, monitoring, and surveillance of AEFI are still vital [44,45]. In addition, different combinations of vaccines have to be trialed to identify the best combination for providing protection against COVID-19, and this can assist researchers in developing an appropriate benefit–risk profile of the vaccines [46]. A heterologous booster vaccination program can simplify the logistics of managing vaccines, help cope with the global vaccine deficiency, and can prolong human cellular immunity and humoral response time against the SARS-CoV-2 variants. The present study has provided information to help inform the general public with respect to making decisions regarding whether to receive a booster vaccine.

This study has several limitations. Firstly, the sample size was small and not sufficient to allow inter-group comparisons for the mixed vaccine group. Secondly, in China, the majority of vaccines are based on the use of inactivated COVID-19 (i.e., Vero cells), which explains why there is a much greater number of participants in the Vero cell group compared to the other groups.

## 5. Conclusions

In conclusion, the potential risk of AEFI in the heterologous mixed vaccine booster group is higher than in the homologous Vero cell and homologous CHO cell booster groups, but still at a level which is considered acceptable. Currently, the heterologous booster vaccination program is a beneficial strategy and provides some valuable data to support vaccine development. The risk of AEFI was higher in the doctors, which may be related to them experiencing high stress levels during the COVID-19 epidemic. Government and non-governmental agencies will need to continue to provide social support to ensure both the physical health and mental wellbeing of HCWs. As major AEFI are very rare, booster vaccination should be encouraged, since it helps prevent a potentially deadly illness and increases protective efficacy against symptomatic SARS-CoV-2 infection.

## Figures and Tables

**Figure 1 vaccines-10-01869-f001:**
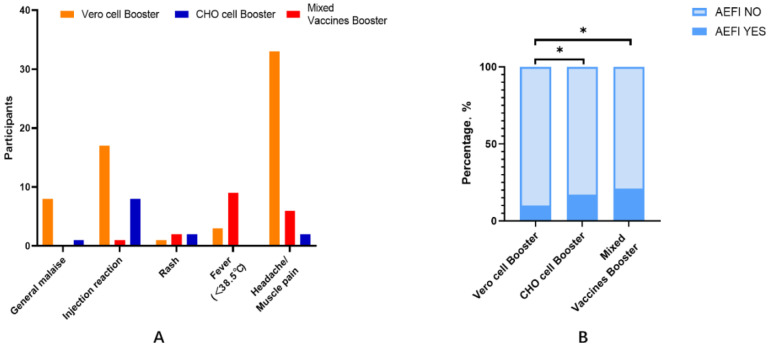
(**A**) AEFI self-reported by participants; (**B**) statistical analysis was performed using the χ² test to compare the rate of AEFI between the three booster vaccine groups. * *p*-value < 0.05 (*p*-value was adjusted by Bonferroni).

**Table 1 vaccines-10-01869-t001:** Characteristics data and need for medical attention after administration of Vero cell booster group, CHO cell booster group, and mixed vaccines booster group.

Characteristics	Vero Cell Booster	CHO Cell Booster	Mixed Vaccines Booster	Total	Statistic	*p*-Value
Age group (in years)						
18–29	368 (58.0%)	61 (81.3%)	38 (46.3%)	467 (59.0%)	χ^2^ = 39.1	<0.001
30–39	163 (25.7%)	6 (8.0%)	17 (20.7%)	186 (23.5%)		
40–49	62 (9.8%)	3 (4%)	10 (12.2%)	75 (9.5%)		
50–60	42 (9.1%)	5 (6.7%)	17 (3.8%)	64 (8.1%)		
Gender					χ^2^ = 12.8	0.002
Female	465 (73.2%)	69 (92.0%)	63 (76.8%)	597 (75.4%)		
Male	170 (26.8%)	6 (8.0%)	19 (23.2%)	195 (24.6%)		
Ethnicity					χ^2^ = 1.689	0.946
Han	403 (63.5%)	51 (67.5%)	53 (66.7%)	507 (64.0%)		
Miao	58 (9.1%)	8 (10.7%)	8 (9.8%)	74 (9.3%)		
Buyi	49 (7.7%)	4 (5.3%)	7 (8.5%)	60 (7.6%)		
Others	125 (19.7%)	12 (16.0%)	14 (17.1%)	151 (19.1%)		
Professional categories					χ^2^ = 23.17	<0.001
Doctors	143 (22.5%)	5 (6.7%)	19 (23.2%)	167 (21.1%)		
Nurses	236 (37.2%)	19 (25.3%)	31 (37.8%)	286 (26.1%)		
Others	256 (40.3%)	51 (68%)	32 (39.0%)	339 (42.8%)		
Level of education					χ^2^ = 25.99	<0.001
Bachelor’s degree or above	349 (55.0%)	18 (24.0%)	40 (48.8%)	407 (51.4%)		
Junior college or below	286 (45.0%)	57 (76.0%)	42 (51.2%)	386 (48.6%)		
Regression of symptoms					Fisher’s exact test	<0.001
Symptomless	573 (90.2%)	62 (82.7%)	65 (79.2%)	700 (88.4%)		
Spontaneous remission	62 (9.8%)	11 (14.7%)	14 (17.1%)	87 (11.0%)		
Seeked help from outpatient provider	0	2 (2.6%)	3 (3.7%)	5 (0.6%)		

**Table 2 vaccines-10-01869-t002:** The risk factors for AEFI following the booster dose of COVID-19 vaccination in HCWs.

Variables	Univariate Analysis		Multivariate Analysis	
	OR (95% CI)	*p*-Value	OR (95% CI)	*p*-Value
Mixed vaccines	2.417 (1.334–4.381)	0.004	2.476 (1.352, 4.533)	0.003
CHO cell booster	1.938 (1.009–3.722)	0.047	2.095 (1.056, 4.157)	0.034
Vero cell booster	Reference		Reference	
Female	2.626 (1.368–5.039)	0.004	2.792 (1.407, 5.543)	0.003
Male	Reference		Reference	
Doctors	2.153 (1.234–3.757)	0.007	2.648 (1.473, 4.760)	0.001
Nurses	1.491 (0.887–2.506)	0.132	1.364 (0.793–2.346)	0.262
Others	Reference		Reference	

Abbreviations: OR, odds ratio; CI, confidence interval; *p*, *p*-value.

## Data Availability

All data in the study are available from the corresponding author by request.

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
