# Peer review of "Investigation of Adverse Events Experienced by Healthcare Workers following Immunization with Homologous or Heterologous COVID-19 Booster Vaccinations"

_vaccines, 2022, doi:10.3390/vaccines10111869_

Round 1

Reviewer 1 Report

The research was adequately designed providing interesting results of certain scientific importance. The manuscript needs an extensive English language revision. Apart from the language revision, it also needs some corrections.

The title is too long (24 words). It should be more concise and simple. The article is about the adverse effects after COVID-19 booster vaccination so the title should address mainly this information excluding the studied group or even the place where the study was done.

The Abstract is too long (462 words) and complicated, not following the journal’s Instructions for Authors.

Inclusion criteria: This part has to be corrected as the inclusion criterion was receiving any type of the vaccine. The division of examinees into three groups is not an inclusion criterion.

Statistical analysis: The significance is addressed only to the statistical significance, not to the results significance.

The part describing why the authors divided HCWs into different categories should not be included in the Patients and Methods part, but should be part of the Introduction or the Discussion paragraph.

Table 1: The meaning of “Statistic” column and values (c2=...) has to be explained in the legend.

“Seventy percent of the health and social workforce are women. Moreover, the health workforce has a vital role in building the resilience of communities and health systems to respond to disasters caused by natural or artificial hazards, as well as related environmental, technological and biological hazards and risks (20).” This text does not discuss the results of the study, and it should be excluded.

“Consequently, the physical and mental health of women may be focused on, and long-term monitoring of women with new vaccination is necessary.” The results of this study do not support this statement.

“The field of PNI is psychoneuroimmunology...” The abbreviation should be defined when appearing in the text for the first time.

The discussion about the connection of stress and the immunity in HCWs should focus on AEFI only.

“In December 2021, the SARS-CoV-2 Omicron variant swiftly overtook the Delta variant to become globally dominant in the COVID-19 pandemic.” Reference is needed for this statement.

Author Response

Dear reviewer,

we are very grateful for your comments on the manuscript. All your comments are very important, they are important for my paper writing and research. According to your advice, we amended the relevant part of the manuscript. Some of your questions were answered below.

  1. The research was adequately designed providing interesting results of certain scientific importance. The manuscript needs an extensive English language revision. Apart from the language revision, it also needs some corrections.

Answer: Thanks so much for your comment. We have already revised the language.

  1. The title is too long (24 words). It should be more concise and simple. The article is about the adverse effects after COVID-19 booster vaccination so the title should address mainly this information excluding the studied group or even the place where the study was done.

Answer: Thanks so much for your comment. We have modified it.

  1. The Abstract is too long (462 words) and complicated, not following the journal’s Instructions for Authors.

Answer: We revised the abstract part according to your suggestion and the word count is 378.

  1. Inclusion criteria: This part has to be corrected as the inclusion criterion was receiving any type of the vaccine. The division of examinees into three groups is not an inclusion criterion.

Answer: Thanks so much for your comment. We have modified it.

  1. Statistical analysis: The significance is addressed only to the statistical significance, not to the results significance.

Answer: We applied Multivariate logistic regression prediction models to find: (i) The risk of AEFI was higher in heterologous vaccines, but Severe adverse events were rare. (ii) Women have a higher risk of AEFI than men; (iii) Doctors showed a higher risk of AEFI in the occupational category.

  1. The part describing why the authors divided HCWs into different categories should not be included in the Patients and Methods part, but should be part of the Introduction or the Discussion paragraph.

Answer: Thanks so much for your comment. We have modified it.

  1. Table 1: The meaning of “Statistic” column and values (c2=...) has to be explained in the legend.

Answer: p-value of the chi-square test or Fisher exact test for all the variables are clearly shown in Table 1. If the same legend is added, we are concerned that it is repetitive with Table 1.

  1. “Seventy percent of the health and social workforce are women. Moreover, the health workforce has a vital role in building the resilience of communities and health systems to respond to disasters caused by natural or artificial hazards, as well as related environmental, technological and biological hazards and risks (20).” This text does not discuss the results of the study, and it should be excluded.

Answer: We have excluded it.

  1. “Consequently, the physical and mental health of women may be focused on, and long-term monitoring of women with new vaccination is necessary.” The results of this study do not support this statement.

Answer: We have deleted it.

  1. “The field of PNI is psychoneuroimmunology...” The abbreviation should be defined when appearing in the text for the first time.

Answer: We have modified it.

  1. The discussion about the connection of stress and the immunity in HCWs should focus on AEFI only.

Answer: Thank you very much for your comment. The reasons for discussing the connection between stress and the immunity in HCWs are as follows: Firstly, data on the most frequent adverse reactions and associated risk factors in HCWs seem necessary to provide more information and increase confidence in these populations in view of a third dose of the vaccine, which has already been implemented. Secondly, our findings showed that doctors are at higher risk of AEFI. According to the reported studies, our analysis may be related to psychoneuroimmunology. At the beginning of the epidemic of SARS-CoV-2 (2019–2020), the Guizhou Provincial Staff Hospital was designated by the local government as a special medical institution for the treatment of COVID-19 in Guizhou Province. HCWs have worked at the frontline of the response to COVID-19. During the first outbreak when the lack of knowledge regarding COVID-19, at that time made HCWs highly vulnerable to stress. HCWs have different pressures depending on their occupational categories. Moreover, studies have shown stress is capable of leading to a host of immune changes involving virtually every aspect of the immune response. And communication between the brain and the immune system is a two-way street. Especially, smaller changes in peripheral inflammation induced by vaccination also lead to changes in symptoms and underlying neural activity. Therefore, we reasoned that the risk of AEFI was higher in doctors which may be related to the high-stress perception during the COVID-19 epidemic. Finally, we appreciate your advice, which is very instructive for our next study. We need well-documented changes in reproductive endocrine, neurological, and mental health status in women and men to explore whether they are relevant to vaccination.

  1. “In December 2021, the SARS-CoV-2 Omicron variant swiftly overtook the Delta variant to become globally dominant in the COVID-19 pandemic.” Reference is needed for this statement.

Answer: Reference has been added.

Hope to get your recognition. Thank you again for your advice. I hope I can learn more from you.

With best regards,

Yaying Li, MD, PhD

Reviewer 2 Report

This objective and outcome of this study are of very little, if any, value to the scientific field of vaccinology, and specifically to the COVID vaccine field.  That is, the objective, if I read it correctly, was to investigate whether "adverse events" (i.e., AE's commonly found after any vaccination dose, including in clinical trials) differentially occur after different vaccine booster doses in health care workers as a specially recognized psychologically challenged group with respect to the COVID pandemic. The authors found, via questionnaires, between 7% and 20% AE's within 3 days after the booster injection in which subunit vaccination led to slightly more AEs than inactivated viral vaccination, and that the "mixed" booster showed the greatest AE rate (i.e., 20%).  

Major issue:  it is well known that such "transient AEs" described in this report, such as general malaise, injection site redness/pain, headache, are very common in the majority of subjects in any country receiving almost any vaccine and is especially well-documented during COVID vaccine clinical trials.  Therefore, I fail to see the value in emphasizing such minor side-effects after receiving different kinds of booster vaccines and am frankly mystified of why the authors conclude the necessity of identifying such AEs in "doctors", of all people, of whom would need specialized social worker aid to console them when/after experiencing such AEs. 

This study could be improved if the investigators were focused on real long-term negative side-effects of such vaccinations (including boosters), such as the well-documented female and male reproductive-endocrine, neurological, and cardiac severities.  

Author Response

Dear reviewer,

we are very grateful for your comments on the manuscript. All your comments are very important, they are important for my paper writing and research. According to your advice, we amended the relevant part of the manuscript. Some of your questions were answered below.

“This objective and outcome of this study are of very little, if any, value to the scientific field of vaccinology, and specifically to the COVID vaccine field.  That is, the objective, if I read it correctly, was to investigate whether "adverse events" (i.e., AE's commonly found after any vaccination dose, including in clinical trials) differentially occur after different vaccine booster doses in healthcare workers as a specially recognized psychologically challenged group with respect to the COVID pandemic. The authors found, via questionnaires, between 7% and 20% AE's within 3 days after the booster injection in which subunit vaccination led to slightly more AEs than inactivated viral vaccination, and that the "mixed" booster showed the greatest AE rate (i.e., 20%).  ”

Answer: The results are not a novelty but prove the safety of the vaccines. Currently, the virus is constantly mutating during the global COVID pandemic, but the number of deaths is falling, proving the effectiveness of the vaccine. Our study provided complementary data on the safety of vaccines. Even though the AEFI of heterologous vaccines is higher than the homologous vaccines, there are no rare or serious adverse reactions. Moreover, heterologous vaccines were reported to be more immunogenic in clinical trials. Therefore, these data should guide future strategies for recording adverse events and future research on COVID-19 vaccination safety.

  1. It is well known that such "transient AEs" described in this report, such as general malaise, injection site redness/pain, and headache, are very common in the majority of subjects in any country receiving almost any vaccine and are especially well-documented during COVID vaccine clinical trials.  Therefore, I fail to see the value in emphasizing such minor side-effects after receiving different kinds of booster vaccines and am frankly mystified of why the authors conclude the necessity of identifying such AEs in "doctors", of all people, of whom would need specialized social worker aid to console them when/after experiencing such AEs. 

Answer: As you said, although clinical trials have documented common adverse effects, detailed surveillance of the adverse reactions in different settings is needed to provide further useful information to clinicians and the public. Moreover, our findings showed that doctors are at higher risk of AEFI. According to the reported studies, our analysis may be related to psychoneuroimmunology. At the beginning of the epidemic of SARS-CoV-2 (2019–2020), the Guizhou Provincial Staff Hospital was designated by the local government as a special medical institution for the treatment of COVID-19 in Guizhou Province. HCWs have worked at the frontline of the response to COVID-19. During the first outbreak when the lack of knowledge regarding COVID-19, at that time made HCWs highly vulnerable to stress. HCWs have different pressures depending on their occupational categories. Moreover, studies have shown stress is capable of leading to a host of immune changes involving virtually every aspect of the immune response. And communication between the brain and the immune system is a two-way street. Especially, smaller changes in peripheral inflammation induced by vaccination also lead to changes in symptoms and underlying neural activity. For the above reasons, we reasoned that the risk of AEFI was higher in doctors which may be related to the high-stress perception during the COVID-19 epidemic. Therefore, we suggested that government and non-governmental agencies should provide social support to focus on the physical and mental health of HCWs.

  1. This study could be improved if the investigators were focused on real long-term negative side-effects of such vaccinations (including boosters), such as the well-documented female and male reproductive-endocrine, neurological, and cardiac severities. 

Answer: We are strongly in favor of your proposal. We appreciate your advice, which is very instructive for our next study.

Hope to get your recognition. Thank you again for your advice. I hope I can learn more from you.

With best regards,

Yaying Li, MD, PhD

Round 2

Reviewer 2 Report

The authors' responses to both concerns of the reviewer that questioned the insignificance of objectives and pretext of the study only strengthened the reviewer's concerns and did not provide an alternative perspective.

Author Response

Dear reviewer,

We are very sorry that the previous response did not address your concerns. Moreover, we greatly appreciate your comments on the manuscript once again.  Some of your questions were answered below sincerely.

Point 1: “This objective and outcome of this study are of very little, if any, value to the scientific field of vaccinology, and specifically to the COVID vaccine field. That is, the objective, if I read it correctly, was to investigate whether "adverse events" differentially occur after different vaccine booster doses in healthcare workers as a specially recognized psychologically challenged group with respect to the COVID pandemic. The authors found, via questionnaires, between 7% and 20% AE's within 3 days after the booster injection in which subunit vaccination led to slightly more AEs than inactivated viral vaccination, and that the "mixed" booster showed the greatest AE rate (i.e., 20%).  ”

Response 1: Thanks so much for your comment. The results are not a novelty but prove the safety of the vaccines. Currently, the virus is constantly mutating during the global COVID pandemic, but the number of deaths is falling, proving the effectiveness of the vaccine. Moreover, the aim of our study was not only to the difference in the occurrence of AEFI but more importantly to analyze the potential risk factors for the occurrence of AEFI after receiving different Booster vaccines. Our study provided complementary data on the safety of booster vaccines. Even though the AEFI of heterologous vaccines is higher than that homologous vaccines, there are no rare or serious adverse reactions. Furthermore, heterologous vaccines were reported to be more immunogenic in clinical trials. Therefore, booster vaccination should be encouraged since it helps prevent a potentially deadly illness and increases protective efficacy against symptomatic SARS-CoV-2 infection. Subsequently, we found that women and doctors were the potential risk factors of AEFI after receiving Booster vaccines via multivariate logistic regression analysis. Hence, these data should guide future strategies for recording adverse events and future research on COVID-19 vaccination safety.

Point 2: It is well known that such "transient AEs" described in this report, such as general malaise, injection site redness/pain, and headache, are very common in the majority of subjects in any country receiving almost any vaccine and are especially well-documented during COVID vaccine clinical trials.  Therefore, I fail to see the value in emphasizing such minor side-effects after receiving different kinds of booster vaccines and am frankly mystified of why the authors conclude the necessity of identifying such AEs in "doctors", of all people, of whom would need specialized social worker aid to console them when/after experiencing such AEs. 

Response 2: Thanks so much for your comment. Although clinical trials have documented common adverse effects, detailed surveillance of the adverse reactions in different settings is needed to provide further useful information to clinicians and the public. Moreover, the best combination of initial and Booster vaccines for protection against COVID-19 is still to be determined. Therefore, the aim of our study that a comparative analysis was performed to investigate the potential risk factors of AEFI after receiving different Booster vaccines among HCWs. 

HCWs have different pressures depending on their role in the workplace. Moreover, The authors of an observational study conducted in Spain reported(i.e. “Factors Associated with Adverse Reactions to BNT162b2 COVID-19 Vaccine in a Cohort of 3969 Hospital Workers”) that women, administrative workers, and workers who had been infected by COVID-19 tended to report more reactions to the vaccine. The authors suggested that men (especially physicians) tend to underreport their symptoms after vaccination, thus biasing the results of pharmacovigilance studies. Therefore, in our study, the occupational categories of HCWs were identified and we found the doctor was a potential risk factor for experiencing AEFI. In the section of the discussion, studies have shown that psychological stress may affect many aspects of the integration of functioning of the immune, central nervous and endocrine systems in both animals and humans. The field of psychoneuroimmunology (PNI) includes studies in which the links between stress and the immune system are examined. A study of PNI showed that stress could induce significant increases in serum IgA, IgG, IgM, C3c, C4, and acute response protein in vivo combined with high-stress perception. These effects extend to vaccine response. Nevertheless, during the COVID-19 pandemic, HCWs were the particularly recognized psychological stress group. Therefore, it is reasonable to suspect that psychological stress may affect the occurrence of AEs. In conclusion, during the COVID-19 pandemic, increasing psychological and social support for HCWs pre-and post-vaccination, and promoting the need for psychological crisis interventions for medical staff, focusing on their physical and mental health, is highly recommended.

Point 3: This study could be improved if the investigators were focused on real long-term negative side-effects of such vaccinations (including boosters), such as the well-documented female and male reproductive-endocrine, neurological, and cardiac severities. 

Response 3: Thanks so much for your comment. We are strongly in favor of your proposal. Unfortunately, during the initial phase of our study, female and male reproductive-endocrine, neurological, and cardiac severities were not well-documented, so that comparative analysis was not performed by long-term follow-up. If the data is supplemented, it will likely cause bias, which may make the results untrue. However, it is necessary to pay more attention to the long-term negative side-effects effects after vaccination. We appreciate your advice, which is very instructive for our next study.

We sincerely hope to get your recognition. Thank you very much for your kind review.

With best regards,

Yaying Li, MD, PhD